# Sparse Multi-Task Reinforcement Learning

**Daniele Calandriello** [*]         **Alessandro Lazaric**[*]         **Marcello Restelli**[†]

Team SequeL                                  DEIB
INRIA Lille – Nord Europe, France            Politecnico di Milano, Italy

## Abstract

In multi-task reinforcement learning (MTRL), the objective is to simultaneously learn multiple tasks and exploit their similarity to improve the performance w.r.t. single-task learning. In this paper we investigate the case when all the tasks can be accurately represented in a linear approximation space using the same small subset of the original (large) set of features. This is equivalent to assuming that the weight vectors of the task value functions are *jointly sparse*, i.e., the set of their non-zero components is small and it is shared across tasks. Building on existing results in multi-task regression, we develop two multi-task extensions of the fitted $Q$-iteration algorithm. While the first algorithm assumes that the tasks are jointly sparse in the given representation, the second one learns a transformation of the features in the attempt of finding a more sparse representation. For both algorithms we provide a sample complexity analysis and numerical simulations.

## 1   Introduction

Reinforcement learning (RL) and approximate dynamic programming (ADP) [24, 2] are effective approaches to solve the problem of decision-making under uncertainty. Nonetheless, they may fail in domains where a relatively small amount of samples can be collected (e.g., in robotics where samples are expensive or in applications where human interaction is required, such as in automated rehabilitation). Fortunately, the lack of samples can be compensated by leveraging on the presence of multiple related tasks (e.g., different users). In this scenario, usually referred to as multi-task reinforcement learning (MTRL), the objective is to simultaneously solve multiple tasks and exploit their similarity to improve the performance w.r.t. single-task learning (we refer to [26] and [15] for a comprehensive review of the more general setting of transfer RL). In this setting, many approaches have been proposed, which mostly differ for the notion of similarity leveraged in the multi-task learning process. In [28] the transition and reward kernels of all the tasks are assumed to be generated from a common distribution and samples from different tasks are used to estimate the generative distribution and, thus, improving the inference on each task. A similar model, but for value functions, is proposed in [16], where the parameters of all the different value functions are assumed to be drawn from a common distribution. In [23] different shaping function approaches for $Q$-table initialization are considered and empirically evaluated, while a model-based approach that estimates statistical information on the distribution of the $Q$-values is proposed in [25]. Similarity at the level of the MDPs is also exploited in [17], where samples are transferred from source to target tasks. Multi-task reinforcement learning approaches have been also applied in partially observable environments [18].

In this paper we investigate the case when all the tasks can be accurately represented in a linear approximation space using the same small subset of the original (large) set of features. This is equivalent to assuming that the weight vectors of the task value functions are *jointly sparse*, i.e., the set of their non-zero components is small and it is shared across tasks. Let us illustrate the concept of shared sparsity using the blackjack card game. The player can rely on a very large number of features such as: value and color of the cards in the player's hand, value and color of the cards on

---

[*]`{daniele.calandriello,alessandro.lazaric}@inria.fr`
[†]`{marcello.restelli}@polimi.it`

the table and/or already discarded, different scoring functions for the player's hand (e.g., sum of the values of the cards) and so on. The more the features, the more likely it is that the corresponding feature space could accurately represent the optimal value function. Nonetheless, depending on the rules of the game (i.e., the reward and dynamics), a very limited subset of features actually contribute to the value of a state and we expect the optimal value function to display a high level of sparsity. Furthermore, if we consider multiple tasks differing for the behavior of the dealer (e.g., the value at which she stays) or slightly different rule sets, we may expect such sparsity to be shared across tasks. For instance, if the game uses an infinite number of decks, features based on the history of the cards played in previous hands have no impact on the optimal policy for any task and the corresponding value functions are all jointly sparse in this representation. Building on this intuition, in this paper we first introduce the notion of sparse MDPs in Section 3. Then we rely on existing results in multi-task regression [19, 1] to develop two multi-task extensions of the fitted $Q$-iteration algorithm (Sections 4 and Section 5) and we study their theoretical and empirical performance (Section 6). An extended description of the results, as well as the full proofs of the statements, are reported in [5].

## 2 Preliminaries

**Multi-Task Reinforcement Learning (MTRL).** A Markov decision process (MDP) is a tuple $\mathcal{M} = (\mathcal{X}, \mathcal{A}, R, P, \gamma)$, where the state space $\mathcal{X}$ is a bounded subset of the Euclidean space, the action space $\mathcal{A}$ is finite (i.e., $|\mathcal{A}| < \infty$), $R : \mathcal{X} \times \mathcal{A} \to [0, 1]$ is the reward of a state-action pair, $P : \mathcal{X} \times \mathcal{A} \to \mathcal{P}(\mathcal{X})$ is the transition distribution over the states achieved by taking an action in a given state, and $\gamma \in (0, 1)$ is a discount factor. A deterministic policy $\pi : \mathcal{X} \to \mathcal{A}$ is a mapping from states to actions. We denote by $\mathcal{B}(\mathcal{X} \times \mathcal{A}; b)$ the set of measurable bounded state-action functions $f : \mathcal{X} \times \mathcal{A} \to [-b, b]$. Solving an MDP corresponds to computing the optimal action–value function $Q^* \in \mathcal{B}(\mathcal{X} \times \mathcal{A}; Q_{\max} = 1/(1-\gamma))$, defined as the fixed point of the optimal Bellman operator $\mathcal{T}$ defined as $\mathcal{T}Q(x, a) = R(x, a) + \gamma \sum_y P(y|x, a) \max_{a'} Q(y, a')$. The optimal policy is obtained as the greedy policy w.r.t. the optimal value function as $\pi^*(x) = \arg\max_{a \in \mathcal{A}} Q^*(x, a)$. In this paper we study the multi-task reinforcement learning (MTRL) setting where the objective is to solve $T$ tasks, defined as $\mathcal{M}_t = (\mathcal{X}, \mathcal{A}, P_t, R_t, \gamma)$ with $t \in [T] = \{1, \dots, T\}$, with the same state-action space, but different dynamics and rewards. The objective of MTRL is to exploit similarities between tasks to improve the performance w.r.t. single-task learning. In particular, we choose linear fitted $Q$-iteration as the single-task baseline and we propose multi-task extensions tailored to exploit the sparsity in the structure of the tasks.

**Linear Fitted Q-iteration.** Whenever $\mathcal{X}$ and $\mathcal{A}$ are large or continuous, we need to resort to approximation schemes to learn a near-optimal policy. One of the most popular ADP methods is the fitted-$Q$ iteration (F$Q$I) algorithm [7], which extends value iteration to approximate action-value functions. While exact value iteration proceeds by iterative applications of the Bellman operator (i.e., $Q^k = \mathcal{T}Q^{k-1}$), at each iteration F$Q$I approximates $\mathcal{T}Q^{k-1}$ by solving a regression problem. Among possible instances, here we focus on a specific implementation of F$Q$I in the fixed design setting with

---

**input:** Input sets $\left\{\mathcal{S}_t = \{x_i\}_{i=1}^{n_x}\right\}_{t=1}^{T}$, $tol$, $K$
&emsp;Initialize $W^0 \leftarrow \mathbf{0}$, $k = 0$
&emsp;**do**
&emsp;&emsp;$k \leftarrow k + 1$
&emsp;&emsp;**for** $a \leftarrow 1, \dots, |\mathcal{A}|$ **do**
&emsp;&emsp;&emsp;**for** $t \leftarrow 1, \dots, T$, $i \leftarrow 1, \dots, n_x$ **do**
&emsp;&emsp;&emsp;&emsp;Sample $r_{i,a,t}^k = R_t(x_{i,t}, a)$ and $y_{i,a,t}^k \sim P_t(\cdot|x_{i,t}, a)$
&emsp;&emsp;&emsp;&emsp;Compute $z_{i,a,t}^k = r_{i,a,t}^k + \gamma \max_{a'} \widetilde{Q}_t^k(y_{i,a,t}^k, a')$
&emsp;&emsp;&emsp;**end for**
&emsp;&emsp;&emsp;Build datasets $\mathcal{D}_{a,t}^k = \{(x_{i,t}, a), z_{i,a,t}^k\}_{i=1}^{n_x}$
&emsp;&emsp;&emsp;Compute $\widehat{W}_a^k$ on $\{\mathcal{D}_{a,t}^k\}_{t=1}^{T}$ (see Eqs. 2,5, or 8)
&emsp;&emsp;**end for**
&emsp;**while** $\left(\max_a \left\|W_a^k - W_a^{k-1}\right\|_2 \geq tol\right)$ **and** $k < K$

Figure 1: Linear F$Q$I with fixed design and fresh samples at each iteration in a multi-task setting.

---

linear approximation and we assume access to a generative model of the MDP. Since the action space $\mathcal{A}$ is finite, we represent action-value functions as a collection of $|\mathcal{A}|$ independent state-value functions. We introduce a $d_x$-dimensional state-feature vector $\phi(\cdot) = [\varphi_1(\cdot), \dots, \varphi_{d_x}(\cdot)]^\mathsf{T}$ with $\phi_i : \mathcal{X} \to \mathbb{R}$ such that $\sup_x ||\phi(x)||_2 \leq L$. From $\phi$ we obtain a linear approximation space for action-value functions as $\mathcal{F} = \{f_w(x, a) = \phi(x)^\mathsf{T} w_a, x \in \mathcal{X}, a \in \mathcal{A}, w_a \in \mathbb{R}^{d_x}\}$. F$Q$I receives as input a fixed set of states $\mathcal{S} = \{x_i\}_{i=1}^{n_x}$ (*fixed design setting*) and the space $\mathcal{F}$. Starting from $w^0 = \mathbf{0}$, at each iteration $k$, F$Q$I first draws a (fresh) set of samples $(r_{i,a}^k, y_{i,a}^k)_{i=1}^{n_x}$ from the generative model of the MDP for each action $a$ on each of the states $\{x_i\}_{i=1}^{n_x}$ (i.e., $r_{i,a}^k = R(x_i, a)$ and $y_{i,a}^k \sim P(\cdot|x_i, a)$) and builds $|\mathcal{A}|$ independent training sets $\mathcal{D}_a^k = \{(x_i, a), z_{i,a}^k\}_{i=1}^{n_x}$, where $z_{i,a}^k = r_{i,a}^k + \gamma \max_{a'} \widehat{Q}^{k-1}(y_{i,a}^k, a')$ is an unbiased sample of $\mathcal{T}\widehat{Q}^{k-1}$ and $\widehat{Q}^{k-1}(y_{i,a}^k, a')$ is com-

puted using the weight vector learned at the previous iteration as $\psi(y_{i,a}^k, a')^\mathsf{T} w^{k-1}$. Then FQI solves $|\mathcal{A}|$ linear regression problems, each fitting the training set $\mathcal{D}_a^k$ and it returns vectors $\widehat{w}_a^k$, which lead to the new action value function $f_{\widehat{w}^k}$ with $\widehat{w}^k = [\widehat{w}_1^k, \ldots, \widehat{w}_{|\mathcal{A}|}^k]$. At each iteration the total number of samples is $n = |\mathcal{A}| \times n_x$. The process is repeated up to $K$ iterations or until no significant change in the weight vector is observed. Since in principle $\widehat{Q}^{k-1}$ could be unbounded (due to numerical issues in the regression step), in computing the samples $z_{i,a}^k$ we use a function $\widetilde{Q}^{k-1}$ obtained by truncating $\widehat{Q}^{k-1}$ in $[-Q_{\max}; Q_{\max}]$. The convergence and the performance of FQI are studied in detail in [20] in the case of bounded approximation space, while linear FQI is studied in [17, Thm. 5] and [22, Lemma 5]. When moving to the multi-task setting, we consider different state sets $\{\mathcal{S}_t\}_{t=1}^T$ and we denote by $\widehat{W}_a^k \in \mathbb{R}^{d_x \times T}$ the matrix with vector $\widehat{w}_{a,t}^k \in \mathbb{R}^{d_x}$ as the $t$–th column. The general structure of FQI in a multi-task setting is reported in Fig. 1. Finally, we introduce the following matrix notation. For any matrix $W \in \mathbb{R}^{d \times T}$, $[W]_t \in \mathbb{R}^d$ is the $t$–th column and $[W]^i \in \mathbb{R}^T$ the $i$–th row of the matrix, $\mathrm{Vec}(W)$ is the $\mathbb{R}^{dT}$ vector obtained by stacking the columns of the matrix, $\mathrm{Col}(W)$ is its column-space and $\mathrm{Row}(W)$ is its row-space. Beside the $\ell_2$, $\ell_1$-norm for vectors, we use the trace (or nuclear) norm $\|W\|_* = \mathrm{trace}((WW^\mathsf{T})^{1/2})$, the Frobenius norm $\|W\|_F = (\sum_{i,j}[W]_{i,j}^2)^{1/2}$ and the $\ell_{2,1}$-norm $\|W\|_{2,1} = \sum_{i=1}^d \|[W]^i\|_2$. We denote by $\boldsymbol{O}^d$ the set of orthonormal matrices and for any pair of matrices $V$ and $W$, $V \perp \mathrm{Row}(W)$ denotes the orthogonality between the spaces spanned by the two matrices.

## 3 Fitted Q–Iteration in Sparse MDPs

Depending on the regression algorithm employed at each iteration, FQI can be designed to take advantage of different characteristics of the functions at hand, such as smoothness ($\ell_2$–regularization) and sparsity ($\ell_1$–regularization). In this section we consider the high–dimensional regression scenario and we study the performance of FQI under sparsity assumptions. Let $\pi_w(x) = \arg\max_a f_w(x, a)$ be the greedy policy w.r.t. $f_w$. We start with the following assumption.[1]

**Assumption 1.** *For any function $f_w \in \mathcal{F}$, the Bellman operator $\mathcal{T}$ can be expressed as*

$$\mathcal{T}f_w(x, a) = R(x, a) + \gamma \mathop{\mathbb{E}}_{x' \sim P(\cdot|x,a)} [f_w(x', \pi_w(x'))] = \psi(x, a)^\mathsf{T} w^R + \gamma \psi(x, a)^\mathsf{T} P_\psi^{\pi_w} w \quad (1)$$

This assumption implies that $\mathcal{F}$ is closed w.r.t. the Bellman operator, since for any $f_w$, its image $\mathcal{T}f_w$ can be computed as the product between features $\psi(\cdot, \cdot)$ and a vector of weights $w^R$ and $P_\psi^{\pi_w} w$. As a result, the optimal value function $Q^*$ itself belongs to $\mathcal{F}$ and it can be computed as $\psi(x, a)^\mathsf{T} w^*$. This assumption encodes the intuition that in the high–dimensional feature space $\mathcal{F}$ induced by $\psi$, the transition kernel $P$, and therefore the system dynamics, can be expressed as a linear combination of the features using the matrix $P_\psi^{\pi_w}$, which depends on both function $f_w$ and features $\psi$. This condition is usually satisfied whenever the space $\mathcal{F}$ is spanned by a very large set of features that allows it to approximate a wide range of different functions, including the reward and transition kernel. Under this assumption, at each iteration $k$ of FQI, there exists a weight vector $w^k$ such that $\mathcal{T}\widehat{Q}^{k-1} = f_{w^k}$ and an approximation of the target function $f_{w^k}$ can be obtained by solving an ordinary least-squares problem on the samples in $\mathcal{D}_a^k$. Unfortunately, it is well known that OLS fails whenever the number of samples is not sufficient w.r.t. the number of features (i.e., $d > n$). For this reason, Asm. 1 is often joined together with a sparsity assumption. Let $J(w) = \{i = 1, \ldots, d : w_i \neq 0\}$ be the set of $s$ non-zero components of vector $w$ (i.e., $s = |J(w)|$) and $J^c(w)$ be the complementary set. In supervised learning, the LASSO [11, 4] is effective in exploiting the sparsity assumption that $s \ll d$ and dramatically reduces the sample complexity. In RL the idea of sparsity has been successfully integrated into policy evaluation [14, 21, 8, 12] but rarely in the full policy iteration. In value iteration, it can be easily integrated in FQI by approximating the target weight vector $w_a^k$ as

$$\widehat{w}_a^k = \arg\min_{w \in \mathbb{R}^{d_x}} \frac{1}{n_x} \sum_{i=1}^{n_x} \left( \phi(x_i)^\mathsf{T} w - z_{i,a}^k \right)^2 + \lambda \|w\|_1. \quad (2)$$

While this integration is technically simple, the conditions on the MDP structure that imply sparsity in the value functions are not fully understood. In fact, one may simply assume that $Q^*$ is sparse in $\mathcal{F}$, with $s$ non-zero weights, thus implying that $d - s$ features captures aspects of states and actions that do not have any impact on the actual optimal value function. Nonetheless, this would provide

no guarantee about the actual level of sparsity encountered by FQI through iterations, where the target functions $f_{w^k}$ may not be sparse at all. For this reason we need stronger conditions on the structure of the MDP. We state the following assumption (see [10, 6] for similar conditions).

**Assumption 2** (Sparse MDPs). *There exists a set $J$ (the set of useful features) for MDP $\mathcal{M}$, with $|J| = s \ll d$, such that for any $i \notin J$, and any policy $\pi$ the rows $[P_\psi^\pi]^i$ are equal to 0, and there exists a function $f_{w^R} = R$ such that $J(w^R) \subseteq J$.*

This assumption implies that not only the reward function is sparse, but also that the features that are useless for the reward have no impact on the dynamics of the system. Since $P_\psi^\pi$ can be seen as a linear representation of the transition kernel embedded in the high-dimensional space $\mathcal{F}$, this assumption corresponds to imposing that the matrix $P_\psi^\pi$ has all its rows corresponding to features outside of $J$ set to 0. This in turn means that the future state-action vector $\mathbb{E}[\psi(x', a')^\mathsf{T}] = \psi(x, a)^\mathsf{T} P_\psi^\pi$ depends only on the features in $J$. In the blackjack scenario illustrated in the introduction, this assumption is verified by features related to the history of the cards played so far. In fact, if we consider an infinite number of decks, the feature indicating whether an ace has already been played is not used in the definition of the reward function and it is completely unrelated to the other features and, thus it does not contribute to the optimal value function. An important consideration on this assumption can be derived by a closer look to the sparsity pattern of the matrix $P_\psi^\pi$. Since the sparsity is required at the level of the rows, this does not mean that the features that do not belong to $J$ have to be equal to 0 after each transition. Instead, their value will be governed simply by the interaction with the features in $J$. This means that the features outside of $J$ can vary from completely unnecessary features with no dynamics, to features that are redundant to those in $J$ in describing the evolution of the system. Additional discussion on this assumption is available in [5]. Assumption 2, together with Asm. 1, leads to the following lemma.

**Lemma 1.** *Under Assumptions 1 and 2, the application of the Bellman operator $\mathcal{T}$ to any function $f_w \in \mathcal{F}$, produces a function $f_{w'} = \mathcal{T} f_w \in \mathcal{F}$ such that $J(w') \subseteq J$.*

This lemma guarantees that at any iteration $k$ of FQI, the target function $f_{w^k} = \mathcal{T}\widehat{Q}^{k-1}$ has a level of sparsity $J(w^k) \leq s$. We are now ready to study the performance of LASSO-FQI over iterations. In order to simplify the comparison to the multi-task results in sections 4 and 5, we analyze the average performance over multiple tasks. We consider that the previous assumptions extend to all the MDPs $\{\mathcal{M}_t\}_{t=1}^T$, each with a set of useful features $J_t$ and sparsity $s_t$. The action–value function learned after $K$ iterations is evaluated by comparing the performance of the corresponding greedy policy $\pi_t^K(x) = \arg\max_a Q_t^K(x, a)$ to the optimal policy. The performance loss is measured w.r.t. a target distribution $\mu \in \mathcal{P}(\mathcal{X} \times \mathcal{A})$. We introduce the following standard assumption for LASSO [3].

**Assumption 3** (Restricted Eigenvalues (RE)). *Define $n$ as the number of samples, and $J^c$ as the complement of the set of indices $J$. For any $s \in [d]$, there exists $\kappa(s) \in \mathbb{R}^+$ such that:*

$$\min\left\{ \frac{\|\Phi\Delta\|_2}{\sqrt{n}\|\Delta_J\|_2} : |J| \leq s, \Delta \in \mathbb{R}^d\backslash\{\mathbf{0}\}, \|\Delta_{J^c}\|_1 \leq 3\|\Delta_J\|_1 \right\} \geq \kappa(s), \tag{3}$$

**Theorem 1** (LASSO-FQI). *Let the tasks $\{\mathcal{M}_t\}_{t=1}^T$ and the function space $\mathcal{F}$ satisfy assumptions 1, 2 and 3 with average sparsity $\bar{s} = \sum_t s_t/T$, $\kappa_{\min}(\bar{s}) = \min_t \kappa(s_t)$ and features bounded $\sup_x \|\phi(x)\|_2 \leq L$. If LASSO-FQI (Alg. 1 with Eq. 2) is run independently on all $T$ tasks for $K$ iterations with a regularizer $\lambda = \delta Q_{\max}\sqrt{\log(d)/n}$, for any numerical constant $\delta > 8$, then with probability at least $(1 - 2d^{1-\delta/8})^{KT}$, the performance loss is bounded as*

$$\frac{1}{T}\sum_{t=1}^T \left\|Q_t^* - Q_t^{\pi_t^K}\right\|_{2,\mu}^2 \leq \mathcal{O}\left(\frac{1}{(1-\gamma)^4}\left[\frac{Q_{\max}^2 L^2}{\kappa_{\min}^4(\bar{s})}\frac{\bar{s}\log d}{n} + \gamma^K Q_{\max}^2\right]\right). \tag{4}$$

**Remark 1 (assumptions).** Asm. 3 is a relatively weak constraint on the representation capability of the data. The RE assumption is common in regression, and it is extensively analyzed in [27]. Asm. 1 and 2 are specific to our setting and may pose significant constraints on the set of MDPs of interest. Asm. 1 is introduced to give a more explicit interpretation for the notion of sparse MDPs. Without Asm. 1, the bound in Eq. 4 would have an additional approximation error term similar to standard approximate value iteration results (see e.g., [20]). Asm. 2 is a potentially very loose sufficient condition to guarantee that the target functions encountered over the iterations of LASSO–FQI have

a minimum level of sparsity. Thm. 1 requires that for any $k \leq K$, the target function $f_{w_t^{k+1}} = \mathcal{T} f_{w_t^k}$ has weights $w_t^{k+1}$ that are sparse, i.e., $\max_{t,k} s_t^k \leq s$ with $s_t^k = |J(w_t^{k+1})|$. In other words, all target functions encountered must be sparse, or LASSO–F$Q$I could suffer a huge loss at an intermediate step. Such condition could be obtained under much less restrictive assumptions than Asm. 2, that leaves up to the MDPs dynamics to resparsify the target function at each step, at the expenses of interpretability. It could be sufficient to prove that the MDP dynamics do not enforce sparsity, but simply do not reduce it across iterations, and use guarantees for LASSO reconstruction to maintain sparsity across iterations. Finally, we point out that even if "useless" features do not satisfy Asm. 2 and are weakly correlated with the dynamics and the reward function, their weights are discounted by $\gamma$ at each step. As a result, the target functions could become "approximately" as sparse as $Q^*$ over iterations, and provide enough guarantees to be used for a variation of Thm. 1. We leave for future work a more thorough investigation of these possible relaxations.

## 4 Group-LASSO Fitted Q–Iteration

After introducing the concept of sparse MDP in Sect. 3, we move to the multi-task scenario and we study the setting where there exists a suitable representation (i.e., set of features) under which all the tasks can be solved using roughly the same set of features, the so-called *shared sparsity* assumption. Given the set of useful features $J_t$ for task $t$, we denote by $J = \cup_{t=1}^T J_t$ the union of all the non-zero coefficients across all the tasks. Similar to Asm. 2 and Lemma 1, we first assume that the set of features "useful" for at least one of the tasks is relatively small compared to $d$ and then show how this propagates through iterations.

**Assumption 4.** *We assume that the joint useful features over all the tasks are such that $|J| = \tilde{s} \ll d$.*

**Lemma 2.** *Under Asm. 2 and 4, at any iteration $k$, the target weight matrix $W^k$ has $J(W^k) \leq \tilde{s}$.*

**The Algorithm.** In order to exploit the similarity across tasks stated in Asm. 4, we resort to the Group LASSO (GL) algorithm [11, 19], which defines a joint optimization problem over all the tasks. GL is based on the intuition that given the weight matrix $W \in \mathbb{R}^{d \times T}$, the norm $\|W\|_{2,1}$ measures the level of shared-sparsity across tasks. In fact, in $\|W\|_{2,1}$ the $\ell_2$-norm measures the "relevance" of feature $i$ across tasks, while the $\ell_1$-norm "counts" the total number of relevant features, which we expect to be small in agreement with Asm. 4. Building on this intuition, we define the GL–F$Q$I algorithm in which at each iteration for each action $a \in \mathcal{A}$ we compute (details about the implementation of GL–F$Q$I are reported in [5, Appendix A])

$$\widehat{W}_a^k = \arg\min_{W_a} \sum_{t=1}^T \left\| Z_{a,t}^k - \Phi_t w_{a,t} \right\|_2^2 + \lambda \left\| W_a \right\|_{2,1}. \tag{5}$$

**Theoretical Analysis.** The regularization of GL–F$Q$I is designed to take advantage of the shared-sparsity assumption at each iteration and in this section we show that this may lead to reduce the sample complexity w.r.t. using LASSO in F$Q$I for each task separately. Before reporting the analysis of GL–F$Q$I, we need to introduce a technical assumption defined in [19] for GL.

**Assumption 5** (Multi-Task Restricted Eigenvalues). *Define $\Phi$ as the block diagonal matrix composed by the $T$ sample matrices $\Phi_t$. For any $s \in [d]$, there exists $\kappa(s) \in \mathbb{R}^+$ s.t.*

$$\min\left\{ \frac{\|\Phi \operatorname{Vec}(\Delta)\|_2}{\sqrt{nT} \|\operatorname{Vec}(\Delta_J)\|_2} : |J| \leq s, \Delta \in \mathbb{R}^{d \times T} \backslash \{\mathbf{0}\}, \|\Delta_{J^c}\|_{2,1} \leq 3 \|\Delta_J\|_{2,1} \right\} \geq \kappa(s), \tag{6}$$

Similar to Theorem 1 we evaluate the performance of GL–F$Q$I as the performance loss of the returned policy w.r.t. the optimal policy and we obtain the following performance guarantee.

**Theorem 2** (GL–F$Q$I). *Let the tasks $\{\mathcal{M}_t\}_{t=1}^T$ and the function space $\mathcal{F}$ satisfy assumptions 1, 2, 4, and 5 with joint sparsity $\tilde{s}$ and features bounded $\sup_x \|\phi(x)\|_2 \leq L$. If GL–F$Q$I (Alg. 1 with Eq. 5) is run jointly on all $T$ tasks for $K$ iterations with a regularizer $\lambda = \frac{L Q_{\max}}{\sqrt{nT}} \left(1 + \frac{(\log d)^{\frac{3}{2}+\delta}}{\sqrt{T}}\right)^{\frac{1}{2}}$, for any numerical constant $\delta > 0$, then with probability at least $(1 - \log(d)^{-\delta})^K$, the performance loss is bounded as*

$$\frac{1}{T} \sum_{t=1}^T \left\| Q_t^* - Q_t^{\pi_t^K} \right\|_{2,\mu}^2 \leq \mathcal{O}\left( \frac{1}{(1-\gamma)^4} \left[ \frac{L^2 Q_{\max}^2}{\kappa^4(2\tilde{s})} \frac{\tilde{s}}{n} \left( 1 + \frac{(\log d)^{3/2+\delta}}{\sqrt{T}} \right) + \gamma^K Q_{\max}^2 \right] \right). \tag{7}$$

**Remark 2 (comparison with LASSO-FQI).** Ignoring all the terms in common with the two methods, constants, and logarithmic factors, we can summarize their bounds of LASSO-F$Q$I and GL–F$Q$I as $\widetilde{\mathcal{O}}(\bar{s}\log(d)/n)$ and $\widetilde{\mathcal{O}}\big(\tilde{s}/n(1+\log(d)/\sqrt{T})\big)$. The first interesting aspect of the bound of GL–F$Q$I is the role played by the number of tasks $T$. In LASSO–F$Q$I the "cost" of discovering the $s_t$ useful features is a factor $\log d$, while GL–F$Q$I has a factor $1+\log(d)/\sqrt{T}$, which decreases with the number of tasks. This illustrates the advantage of the multi–task learning dimension of GL–F$Q$I, where all the samples of all tasks actually contribute to discovering useful features, so that the more the number of features, the smaller the cost. In the limit, we notice that when $T \to \infty$, the bound for GL–F$Q$I does not depend on the dimensionality of the problem anymore. The other critical aspect of the bounds is the difference between $\bar{s}$ and $\tilde{s}$. In fact, $\max_t s_t \leq \tilde{s} \leq d$ and if the shared-sparsity assumption does not hold, we can construct cases where the number of non-zero features $s_t$ is very small for each task, but the *union $J = \cup_t J_t$* is still a full set, so that $\tilde{s} \approx d$. In this case, GL–F$Q$I cannot leverage on the shared sparsity across tasks and it may perform significantly worse than LASSO–F$Q$I. This is the well–known *negative transfer* effect that happens whenever the wrong assumption over tasks is enforced thus worsening the single-task learning performance.

## 5  Feature Learning Fitted Q–Iteration

Unlike other properties such as smoothness, the sparsity of a function is intrinsically related to the specific *representation* used to approximate it (i.e., the function space $\mathcal{F}$). While Asm. 2 guarantees that $\mathcal{F}$ induces sparsity for each task separately, Asm. 4 requires that all the tasks share the same useful features in the given representation. As discussed in Rem. 2, whenever this is not the case, GL–F$Q$I may perform worse than LASSO–F$Q$I. In this section we investigate an alternative notion of sparsity in MDPs and we introduce the Feature Learning fitted Q-iteration (FL–F$Q$I) algorithm.

**Low Rank approximation.** Since the poor performance of GL–F$Q$I is due to the chosen representation (i.e., features), it is natural to ask the question whether there exists an alternative representation (i.e., different features) inducing a higher level of shared sparsity. Let us assume that there exists a space $\mathcal{F}^*$ defined by features $\phi^*$ such that the weight matrix of the optimal Q-functions $A^* \in \mathbb{R}^{d \times T}$ is such that $J(A^*) = s^* \ll d$. As shown in Lemma 2, together with Asm. 2 and 4, this guarantees that at any iteration $J(A^k) \leq s^*$. Given the set of states $\{\mathcal{S}_t\}_{t=1}^T$, let $\Phi$ and $\Phi^*$ the feature matrices obtained by evaluating $\phi$ and $\phi^*$ on the states. We assume that there exists a linear transformation of the features of $\mathcal{F}^*$ to the features of $\mathcal{F}$ such that $\Phi = \Phi^* U$ with $U \in \mathbb{R}^{d_x \times d_x}$. In this setting the samples used to define the regression problem can be formulated as noisy observations of $\Phi^* A_a^k$ for any action $a$. Together with the transformation $U$, this implies that there exists a weight matrix $W_a^k$ such that $\Phi^* A_a^k = \Phi^* U U^{-1} A_a^k = \Phi W_a^k$ with $W_a^k = U^{-1} A_a^k$. Although $A_a^k$ is indeed sparse, any attempt to learn $W_a^k$ using GL would fail, since $W_a^k$ may have a very low level of sparsity. On the other hand, an algorithm able to learn a suitable transformation $U$, it may be able to recover the representation $\Phi^*$ (and the corresponding space $\mathcal{F}^*$) and exploit the high level of sparsity of $A_a^k$. While this additional step of representation or *feature learning* introduces additional complexity, it allows to relax the strict assumption on the joint sparsity $\tilde{s}$ and may improve the performance of GL–F$Q$I. Our assumption is formulated as follows.

**Assumption 6.** *There exists an orthogonal matrix $U \in \mathbf{O}^d$ (block diagonal matrix having matrices $\{U_a \in \mathbf{O}^{d_x}\}$ on the diagonal) such that the weight matrix $A^*$ obtained as $A^* = U^{-1} W^*$ is jointly sparse, i.e., has a set of "useful" features $J(A^*) = \cup_{t=1}^T J([A^*]_t)$ with $|J(A^*)| = s^* \ll d$.*

Coherently with this assumption, we adapt the multi-task feature learning (MTFL) algorithm defined in [1] and at each iteration $k$ for any action $a$ we solve the optimization problem

$$(\widehat{U}_a^k, \widehat{A}_a^k) = \arg \min_{U_a \in \mathbf{O}^d} \min_{A_a \in \mathbb{R}^{d \times T}} \sum_{t=1}^T ||Z_{a,t}^k - \Phi_t U_a [A_a]_t||^2 + \lambda\, ||A||_{2,1}\,. \tag{8}$$

In order to better characterize the solution to this optimization problem, we study more in detail the relationship between $A^*$ and $W^*$ and analyze the two directions of the equality $A^* = U^{-1} W^*$. When $A^*$ has $s^*$ non-zero rows, then any orthonormal transformation $W^*$ will have at most rank $r^* = s^*$. This suggests that instead of solving the joint optimization problem in Eq. 8 and explicitly recover the transformation $U$, we may directly try to solve for low-rank weight matrices $W$. Then we need to show that a low-rank $W^*$ does indeed imply the existence of a transformation to a jointly-sparse matrix $A^*$. Assume $W^*$ has low rank $r^*$. It is then possible to perform a standard singular

value decomposition $W^* = U\Sigma V = UA^*$. Because $\Sigma$ is diagonal with $r^*$ non-zero entries, $A^*$ will have $r^*$ non-zero rows, thus being jointly sparse. It is possible to derive the following equivalence.

**Proposition 1** ([5, Appendix A])**.** *Given* $A, W \in \mathbb{R}^{d \times T}$, $U \in \mathbf{O}^d$, *the following equality holds, with the relationship between the optimal solutions being* $W^* = UA^*$,

$$\min_{A,U} \sum_{t=1}^{T} ||Z_{a,t}^k - \Phi_t U_a [A_a]_t||^2 + \lambda \, ||A||_{2,1} = \min_{W} \sum_{t=1}^{T} ||Z_{a,t}^k - \Phi_t [W_a]_t||^2 + \lambda ||W||_1. \quad (9)$$

The previous proposition states the equivalence between solving a feature learning version of GL and solving a nuclear norm (or trace norm) regularized problem. This penalty is equivalent to an $\ell_1$-norm penalty on the singular values of the $W$ matrix, thus forcing $W$ to have low rank. Notice that assuming that $W^*$ has low rank can be also interpreted as the fact that either the task weights $[W^*]_t$ or the features weights $[W^*]^i$ are linearly correlated. In the first case, it means that there is a dictionary of core tasks that is able to reproduce all the other tasks as a linear combination. As a result, Assumption 6 can be reformulated as $\mathrm{Rank}(W^*) = s^*$. Building on this intuition we define the FL–F$Q$I algorithm where the regression is carried out according to Eq. 9.

**Theoretical Analysis.** Our aim is to obtain a bound similar to Theorem 2 for the new FL-F$Q$I Algorithm. We begin by introducing a slightly different assumption on the data available for regression.

**Assumption 7** (Restricted Strong Convexity)**.** *Under Assumption 6, let* $W^* = UDV^\mathsf{T}$ *be a singular value decomposition of the optimal matrix* $W^*$ *of rank* $r$, *and* $U^r, V^r$ *the submatrices associated with the top* $r$ *singular values. Define* $\mathcal{B} = \{\Delta \in \mathbb{R}^{d \times T} : \mathrm{Row}(\Delta) \perp U^r$ *and* $\mathrm{Col}(\Delta) \perp V^r\}$, *and the projection operator onto this set* $\Pi_{\mathcal{B}}$. *There exists a positive constant* $\kappa$ *such that*

$$\min \left\{ \frac{\|\Phi \, \mathrm{Vec}(\Delta)\|_2^2}{2nT \| \, \mathrm{Vec}(\Delta)\|_2^2} : \Delta \in \mathbb{R}^{d \times T}, \|\Pi_{\mathcal{B}}(\Delta)\|_1 \leq 3\|\Delta - \Pi_{\mathcal{B}}(\Delta)\|_1 \right\} \geq \kappa \quad (10)$$

**Theorem 3** (FL–F$Q$I)**.** *Let the tasks* $\{\mathcal{M}_t\}_{t=1}^{T}$ *and the function space* $\mathcal{F}$ *satisfy assumptions 1, 2, 6, and 7 with rank* $s^*$, *features bounded* $\sup_x ||\phi(x)||_2 \leq L$ *and* $T > \Omega(\log n)$. *If* FL–F$Q$I *(Alg. 1 with Eq. 8) is run jointly on all* $T$ *tasks for* $K$ *iterations with a regularizer* $\lambda \geq 2LQ_{\max}\sqrt{(d+T)/n}$, *then with probability at least* $\Omega((1 - \exp\{-(d+T)\})^K)$, *the performance loss is bounded as*

$$\frac{1}{T} \sum_{t=1}^{T} \left\| Q_t^* - Q_t^{\pi_t^K} \right\|_{2,\rho}^2 \leq \mathcal{O} \left( \frac{1}{(1-\gamma)^4} \left[ \frac{Q_{\max}^2 L^4}{\kappa^2} \frac{s^*}{n} \left( 1 + \frac{d}{T} \right) + \gamma^K Q_{\max}^2 \right] \right).$$

**Remark 3 (comparison with GL-F$Q$I).** Unlike GL–F$Q$I, the performance FL–F$Q$I does not depend on the shared sparsity $\tilde{s}$ of $W^*$ but on its rank, that is the value $s^*$ of the most jointly-sparse representation that can be obtained through an orthogonal transformation $U$ of the features. Whenever tasks are somehow *linearly dependent*, even if the weight matrix $W^*$ is dense and $\tilde{s} \approx d$, the rank $s^*$ can be small, thus guaranteeing a dramatic improvement over GL–F$Q$I. On the other hand, learning a new representation comes at the cost of a worse dependency on $d$. In fact, the term $\log(d)/\sqrt{T}$ in GL–F$Q$I, becomes $d/T$, implying that many more tasks are needed for FL–F$Q$I to construct a suitable representation. This is not surprising since we introduced a $d \times d$ matrix $U$ in the optimization problem and a larger number of parameters needs to be learned. As a result, although significantly reduced by the use of trace-norm instead of $\ell_{2,1}$-regularization, the negative transfer is not completely removed. In particular, the introduction of new tasks, that are not linear combinations of the previous tasks, may again increase the rank $s^*$, corresponding to the fact that no jointly-sparse representation can be constructed.

## 6 Experiments

We investigate the empirical performance of GL–F$Q$I, and FL–F$Q$I and compare their results to single-task LASSO–F$Q$I in two variants of the blackjack game. In the first variant (*reduced variant*) the player can choose to *hit* to obtain a new card or *stay* to end the episode, while in the second one (*reduced variant*) she can also choose to *double* the bet on the first turn. Different tasks can be defined depending on several parameters of the game, such as the number of decks, the threshold at which the dealer stays and whether she hits when the threshold is research exactly with a *soft* hand.

**Full variant experiment.** The tasks are generated by selecting $2, 4, 6, 8$ decks, by setting the stay threshold at $\{16, 17\}$ and whether the dealer hits on soft, for a total of 16 tasks. We define a very

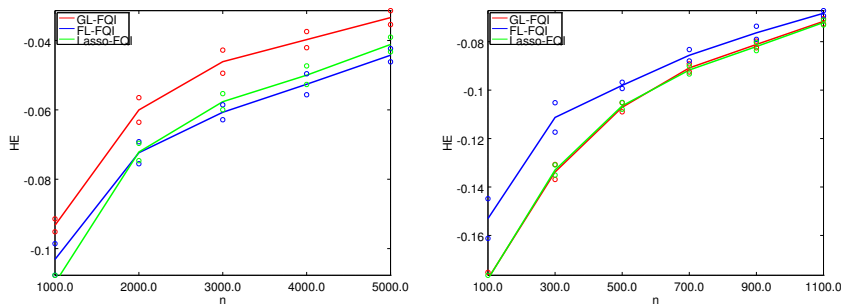

Figure 2: Comparison of FL–F$Q$I, GL–F$Q$I and LASSO–F$Q$I on full *(left)* and reduced *(right)* variants. The $y$ axis is the average house edge (HE) computed across tasks.

rich description of the state space with the objective of satisfying Asm. 1. At the same time this is likely to come with a large number of useless features, which makes it suitable for sparsification. In particular, we include the player hand value, indicator functions for each possible player hand value and dealer hand value, and a large description of the cards not dealt yet (corresponding to the history of the game), under the form of indicator functions for various ranges. In total, the representation contains $d = 212$ features. We notice that although none of the features is completely useless (according to the definition in Asm. 2), the features related with the history of the game are unlikely to be very useful for most of the tasks defined in this experiment. We collect samples from up to 5000 episodes, although they may not be representative enough given the large state space of all possible histories that the player can encounter and the high stochasticity of the game. The evaluation is performed by simulating the learned policy for 2,000,000 episodes and computing the average House Edge (HE) across tasks. For each algorithm we report the performance for the best regularization parameter $\lambda$ in the range $\{2, 5, 10, 20, 50\}$. Results are reported in Fig. 2-(left). Although the set of features is quite large, we notice that all the algorithms succeed in learning a good policy even with relatively few samples, showing that all of them can take advantage of the sparsity of the representation. In particular, GL–F$Q$I exploits the fact that all 16 tasks share the same useless features (although the set of useful feature may not overlap entirely) and its performance is the best. FL–F$Q$I suffers from the increased complexity of representation learning, which in this case does not lead to any benefit since the initial representation is sparse, but it performs as LASSO–F$Q$I.

**Reduced variant experiment.** We consider a representation for which we expect the weight matrix to be dense. In particular, we only consider the value of the player's hand and of the dealer's hand and we generate features as the Cartesian product of these two discrete variables plus a feature indicating whether the hand is soft, for a total of 280 features. Similar to the previous setting, the tasks are generated with $2, 4, 6, 8$ decks, whether the dealer hits on soft, and a larger number of stay thresholds in $\{15, 16, 17, 18\}$, for a total of 32 tasks. We used regularizers in the range $\{0.1, 1, 2, 5, 10\}$. Since the history is not included, the different number of decks influences only the probability distribution of the totals. Moreover, limiting the actions to either *hit* or *stay* further increases the similarity among tasks. Therefore, we expect to be able to find a dense, low-rank solution. Results in Fig. 2-(right) confirms this guess, with FL–F$Q$I performing significantly better than the other methods. In addition, GL–F$Q$I and LASSO–F$Q$I perform similarly, since the dense representation penalizes both single-task and shared sparsity; in fact, both methods favor low values of $\lambda$, meaning that the sparse-inducing penalties are not effective.

## 7 Conclusions

We studied the multi-task reinforcement learning problem under shared sparsity assumptions across the tasks. GL–F$Q$I extends the F$Q$I algorithm by introducing a Group-LASSO step at each iteration and it leverages over the fact that all the tasks are expected to share the same small set of useful features to improve the performance of single-task learning. Whenever the assumption is not valid, GL–F$Q$I may perform worse than LASSO–F$Q$I. With FL–F$Q$I we take a step further and we learn a transformation of the given representation that could guarantee a higher level of shared sparsity. Future work will be focused on considering a relaxation of the theoretical assumptions and on studying alternative multi-task regularization formulations such as in [29] and [13].

**Acknowledgments** This work was supported by the French Ministry of Higher Education and Research, the European Community's Seventh Framework Programme under grant agreement 270327 (project CompLACS), and the French National Research Agency (ANR) under project ExTra-Learn n.ANR-14-CE24-0010-01.

## Footnotes

[1]A similar assumption has been previously used in [9] where the transition $P$ is embedded in a RKHS.

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
