[Reviews · NeurIPS 2014]

Submitted by Assigned_Reviewer_4

This paper focuses on l_1 regularized multi-task feature RL by means of an integration between multi-task feature learning (MTFL) and Fitted Q-learning.

Clarity:
The paper is mostly well written. Regarding the format of this paper, the font size is not right.
A suggestion on nuclear norm: The nuclear norm is usually represented as ||\cdot||_*, where in the paper it is notated as ||\cdot||_1.
There is a mistake in Assumption 5. In line 291, the author claims that $W*= U^{-1}A*$ and later in line 299 the author claims that $A*= w^{-1}W*$. Judging from the context, I think line 291 is right and line 299 is mistakenly written, and thus the formulation in Equation (5, 6) are wrong, where U should be U^{-1}. Such serious notation mistakes should be avoid in NIPS submission.

Quality:
There are some concerns regarding the algorithm which I would like the author to reply:

1. The convergence of the FQI algorithm.
The first concern is that fitted Q is picky on the regressor: it may diverge unless a certain kind of regressor is used such as kernel averaging and tree-based regressor. However, this is not mentioned in the paper. Assume the author uses these two kinds of regressors, it is not quite explicit on how to extend these two regressors to l_1 regularized regression, because these two regressors (non-parametric and tree-based) are known to be difficult to incorporate l_1 regularizations. If other regressor are used, then the convergence of the algorithm is not guaranteed any more.

2. line 314, eqt (6).
This is a critical step in the MTF-RL framework the paper proposed. The reviewer would like to know If the author is able to provide proof on this.
The relaxation from equation (5) to (6) is a critical part, which relaxes group norm to trace norm.
The author should give a rigorous proof on this, although he explains it a little bit line 318-321 and in the Appendix. Since there is no space limitation in the Appendix, the author can write the full deduction there. Intuitively speaking, if A matrix is jointly sparse, then W matrix is a low rank matrix, however, if W is low rank, it can NOT automatically be deducted that A is jointly sparse. So the reviewer would like to know both the proof and explicit explanation (if there is) between the original group lasso objective function and its relaxed problem.

Originality:
MTFL and Fitted-Q has been extensively studied, yet it is interesting to explore the integration of the two.

Significance:
not quite important. Considering my concerns of the paper, the reviewer doubt the effectiveness and soundness of the paper.
Summary: The reviewer has certain doubts about the fundamentals of the algorithm regarding the convergence and feasibility as mentioned in the review opinion. I would like to hear from the author about the reviewer's two concerns, especially the proof of the second concern.

Submitted by Assigned_Reviewer_16

This paper presents an approach for efficient learning in multitask RL settings, where the assumption is made that the value functions across tasks are jointly sparse. The paper first presents an algorithm for fitted Q-iteration across tasks, and then a method for finding linear combinations of existing features that are sparse across tasks.

Broadly speaking, this kind of approach to leveraging structure is important to the field, and will, I think, be critical in scaling things up. I was initially skeptical that this kind of structure would be present in many tasks, but the BlackJack example is reasonably compelling, so I was pleased by the basic direction of the paper. I'd like to see a few more explains to get a feel for where this might reasonably be attempted.

I'd have much preferred a paper where the authors used this kind of idea to solve several interesting problems, and used the resulting sparsity to develop some human-intuitive insight into their domains, but I guess a collection of proofs is OK too.

A small question - surely linear Fitted Q iteration is just LSPI. Yes?

It would be really nice if the authors spent a bit more space explaining what various assumptions mean. How badly are these violated in practice? When, like equation 4, they are listed but then not explained, one gets the nagging worry that they are almost never true, and we are proving theorems for the sake of it. Making them a little more intuitive will help dispel this mistaken impression.

Small writing things:
o The first sentence contains a type error. Reinforcement learning is a setting, or paradigm. Approximate dynamic programming is an approach, or a method, or a family of approaches.
o "leveraging on the presence" -> delete "on"
o Parenthetical citations are not nouns. The rest of the paper is generally well written, but the authors should be sure to use actual nouns as nouns. Technical depth and good ideas are not an excuse to fail at basic grammar.
o "The more the features," -> "The more features"
o "in a given state ," -> "in a given state," (delete space)
o Q_max = 1/(1-\gamma) did I miss something? I think the equals sign is a typo.
o X_t and X_t do not need the subscript t if they are constant wrt t.
o "always in the F" -> "always in F"
o "and it is usually verified whenever the space F", "verified" is a strong word here. "it usually approximately holds"?
o "reduce the sample complexity and improve the average performance" -> delete both "the"s.
o "this may lead to reduce the" -> "this may reduce the"
o "so that the more the number of features, the smaller the cost" -> "so that the more features, the lower the cost"
o "threshold is research" -> "reached"
o Equation 7 needs a period, the one following should have a comma not a period.
o Please check your reference typesetting (e.g., "mdps")
Summary: A potentially interesting paper, though I would've liked to have seen the authors do something more interesting with this idea.

Submitted by Assigned_Reviewer_41

Summary: This paper addresses the problem of multi-task reinforcement learning. The underlying intuition is that the value functions of all tasks can jointly be represented as a sparse combination of given basis functions. The paper introduces two algorithms, which are both variants of the Fitted Q-Iteration algorithm. The difference between these two algorithms is how they solve the regression problem at each iteration of FQI.
To exploit the joint sparsity, Group LASSO Fitted Q-Iteration (GL-FQI) solves a group Lasso regression problem at each iteration. The regularizer in the Group Lasso encourages finding a sparse set of weights that are jointly used in all tasks.
The other algorithm is Feature Learning FQI (FL-FQI). The underlying assumption of FL-FQI is that there exists a linear transformation of the features that make the tasks jointly sparse in the transformed space.
With certain set of assumptions, the paper provides error upper bound for both algorithms.
One may summarize the main finding of the theoretical guarantee for GL-FQI (FL-FQI) as that the joint sparsity level (the rank of weight matrix) appears in the upper bound. Moreover, if the number of tasks (T) is large, the effect of the number of features (d) becomes less important (the behavior would be log(d) / T for GL-FQI and d / T for FL-FQI). This latter point makes these methods different from a single-task Lasso FQI, which has log(d) behavior (no division by T).
The paper also has experimental results comparing LASSO FQI, GL-FQI, and FL-FQI.

This is an interesting paper. It is written clearly and has both theory and experiments (though please refer to my comments below regarding the theory, as some aspects of it are not clear). Benefitting from sparsity is not new in RL (both in the API and AVI contexts), but as far as I know, it is new in the multi-task RL setup. Even though GP-FQI might not seem to be a surprisingly novel algorithm, I found FL-FQI quite compelling as it tries to find a transformation that makes the problem sparse.

Quality: The paper has both theory and experiments. The theory has some simplifying assumptions (especially Assumptions 1 and 2), but the message is clear: for some problems we might benefit from the joint sparsity of value functions. But there are some concerns about the theory, which I describe below.

Clarity: The paper is well-written and well-organized. Some parts of it can be improved though.

Originality: The general structure of the algorithms is not surprisingly novel (in particular, GL-FQI), but as far as I know, they have not been suggested before.

Significance: The algorithms developed in this paper are good additions to the toolbox of RL practitioners.

Detailed Comments (roughly ordered according to their importance):

[Major] Please address these comments.

* In Appendix C, the authors used a result from Lounici et al. 2011. Is it possible to be a bit more specific about the referred result?
If it is Corollary 4.1 in that paper, then their noise model is for Gaussian noise. This is not the case here. Also the constants should be 128/k^2 instead of 1280/k^4.
If it is Theorem 8.1, which is for non-Gaussian case, then the bound would be different (we have (log d)^{3/2 + delta} / sqrt{T} ) behavior instead of log d/T behavior).

* If I understand Assumption 2 correctly, it indicates that the the i-th basis function phi_i is uncorrelated to (P phi_j). Later Lemma 7 shows that this along Assumption 1 imply that the application of the Bellman operator on a function f does not change its sparsity property.

I have two main concerns here:
1) Why the probability distribution for this correlation is chosen to be the Lebesgue measure? Why not mu or rho?
2) In the proof of Lemma 7, w_1 is the weight that leads to a function f_{w_1} that has zero L_2-norm distance from T f_{w_0}. But this does not imply that they are pointwise equal. If my claim is correct and I did not misunderstood anything, isn't it possible that we have a large difference on a set of measure 0 (w.r.t. the Lebesgue measure)?
I believe if we require that the design matrix does not sample from those small sets, the analysis should be fine. Please comment on this issue.

* The value of k(s) in Assumption 4 is random, since the design matrix is random. This makes the upper bounds of both theorems random. Also the work of Lounici et al. 2011 is for fixed design. How does it affect the results?

* One other issue is that you reuse the dataset in all iteration. It makes them dependent.

[Minor] If you have enough space, please address these concerns.

* L118: Assumption 1 is much stronger than requiring that Q* belong to the function space. Is there any specific reason not to study the function approximation error? Is it because Proposition 1 in the supplementary material does not hold?

* L095: Features are defined over the state space, but not state-action space.

* L213: In the definition of Restricted Eigenvalues, should we have |J| \leq s or |J| \leq \tilde{s} ?

* In both Theorems 1 and 2, the probability of the event is written in a non-standard form. Is it possible to simplify it in the usual (eps,delta) form?

* What would happen if instead of l_{2,1}-norm on W, we use l_{1,1}? What is the motivation of using l_2-norm of the rows?

* A short discussion on how to solve Equation (5)?

* A relevant work is recently done by Brunskill and Li, 2013. Please compare with it.

* L220: Is the distribution of samples rho or mu? Based on the definition of the concentrability coefficient, it seems that it should be mu.

* L287: Typo: "Let X the" ---> "Let X be the"

* In Assumption 5, is O^d orthogonal or orthonormal (as suggested on L102)?

- Lounici, Pontil, Van De Geer, Tsybakov, et al, "Oracle inequalities and optimal inference under group sparsity," The Annals of Statistics, 2011.
- Brunskill and Li, "Sample complexity of multi-task reinforcement learning," UAI, 2013.
Summary: This paper considers the problem of multi-task RL and proposes two algorithms to benefit from the joint sparsity of value functions of the tasks. This is an interesting paper, but there are some concerns about its theoretical analysis.
Author Feedback
Author rebuttal: We thank the reviewers for their helpful and thoughtful feedback. Due to the space constraint we only address the main points here.

Rev.1
1.LSPI
Linear FQI and LSPI use a linear architecture to approximate value functions but they differ in being value and policy iteration algorithms.

2.Assumptions
Asms. 5 and 6 are sufficient but not necessary. Intuitively, they measure how much the available samples support the reconstruction of the target function using only a relevant subset of features. In some sense, k(s) plays a role which in other regression schemes is played by the smallest eigenvalue of the design matrix (eg, ordinary least-squares). Other more easily checkable conditions can be considered but they are often too strong (see [van de Geer, Buehlmann. On the conditions used to prove oracle results for the Lasso. 2009]). In the experiments not all the assumptions hold but the statements of the theorems seem to be respected in their dependency on dimensionality, number of samples, and number of tasks. Other assumptions are discussed in Remark 2.

Rev.2
1.Convergence of FQI
Linear FQI with bounded features and truncation at each iteration (we will make these requirements explicit) is proved to converge to a bounded region around the optimal value function, thus it cannot diverge (see [Thm.5 in Sect.B, Lazaric and Restelli, Transfer from Multiple MDPs, NIPS 2011] or [Lem.5, Scherrer et al, Approximate Modified Policy Iteration, ICML 2012]). This is true for any API algorithm using a regression algorithm with bounded per-iteration error (see [Munos, Performance bounds in Lp norm for approximate value iteration, 2007] and [19]). As suggested by the reviewer, when "averagers" are used (eg, normalized RBFs, CMAC, kernel averaging), the joint approximation-Bellman operator is a contraction and the convergence to a point is guaranteed with a potential improvement in performance [Gordon, Stable Function Approximation in Dynamic Programming, ICML 1995], [Ernst et Al., Tree-Based Batch Mode Reinforcement Learning, JMLR 2005]. Empirically, linear regression has been often successfully used in the past [eg, Timmer and Riedmiller, Fitted q iteration with cmacs, ADPRL 2007]. In our experiments, no algorithm displayed any erratic behavior or oscillation.

2.Derivation of eq.6
Technically speaking (6) is not a "relaxation" of (5) but rather an equivalence between two optimization problems. As noticed by the reviewer, Lem.1 in App.A guarantees that when the solution A of (5) is sparse, W is low rank. Intuitively, the converse follows from W=UA (we apologize for the typo in the paper, we will make the notation consistent and use W*=UA*) and the singular value decomposition W = U S V^T, from which A = S V^T, which implies that A is both sparse (since W is low rank) and with all its non-zero rows orthogonal. The global statement is reported in Thm.1 of [1] (referenced in our paper). The proof of (6) is not trivial but it is entirely proved in [1] and we preferred to avoid reproducing the exact same (lengthy) proof in our paper (notice that the change in the definition of the regularizer does not affect the proofs). Roughly speaking, from Lem.1 we have the equivalence (10)=(12), then using Lem.2 and setting epsilon=0, we obtain (12)=(15), hence (6) follows. Referring to the [1], the key statements are in Thm.1,2,3 and Cor.1, which together cover (6), Lem.1 and 2 in our paper. We will make the logic of the proof and references to [1] more explicit and detailed.

Rev.3
1.Gaussian noise
In the paper we have (non-Gaussian) bounded zero-mean noise, so the reviewer is right: we should use Thm.8.1 from [18] rather than Corollary 4.1. As he/she suggests, this changes the final bound as (log d)^{3/2+delta}/sqrt{T}. The high-level properties of the algorithm (as discussed in the remarks) are not affected by this change.

2.Assumption 2
The choice of Lebesgue measure is motivated by the attempt of keeping Asm.2 as general as possible independently from the measures used for sampling (either a distribution over XxA in case of random design or a specific set of inputs in case of fixed-design) and testing. The remark of the reviewer is correct but the set of fixed-design points compromising the use of Asm.2 is indeed a set with zero Lebesgue measure, that could be explicitly excluded from the "feasible" points in the fixed design (we will fix this).

3.Value of k(s)
The current analysis is carried out in fixed design and thus k(s) is not random and the assumption is relative to the specific choice of samples (x_i,a_i). The definition of a sampling distribution (L369) is not necessary and only a testing distribution to measure the performance loss is required. Removing the notion of sampling distribution does not affect Thm.1 since Prop.1 bounds the prediction loss in L-infinity norm (and it would indeed simplify the definition of concentrability terms in Asm.7). We apologize for the confusion, we will clarify this point in the final version.

4.Dependency across iterations
While the case of a fixed single-sample across iterations is addressed in the general case of approximate value iteration in sect.4.2 of [19], here we refresh the samples at each iteration. While the (x_i,a_i) part is fixed-design and so it is constant through iterations, the next states x'_i (and the reward r_i if stochastic) are resampled at each iteration. This avoids correlation across iterations and it allows using a simple union bound over K iterations to prove our statements. We will clarify this, while we leave the analysis of the single-sample case for future work.

5.l_{1,1} norm
l_{1,1} implements a looser notion of "multi-task sparsity" than l_{2,1}. To the best of our knowledge, the theoretical understanding of l_{1,1} is less developed than for l_{2,1} but it is an interesting direction for future work.

6.Solution of eq.5
We have not included a detailed discussion because it mostly follows from [1].